# Endocrinopathies in Hemoglobinopathies: What Is the Role of Iron?

**DOI:** 10.3390/ijms242216263

**Published:** 2023-11-13

**Authors:** Paschalis Evangelidis, Theodora-Maria Venou, Barmpageorgopoulou Fani, Efthymia Vlachaki, Eleni Gavriilaki

**Affiliations:** 1Second Propedeutic Department of Internal Medicine, Hippocration Hospital, Aristotle University of Thessaloniki, 54642 Thessaloniki, Greece; pascevan@auth.gr; 2Adult Thalassemia Unit, 2nd Department of Internal Medicine, Aristotle University of Thessaloniki, Hippocration General Hospital, 54642 Thessaloniki, Greece; theoveno@auth.gr (T.-M.V.); efivlachaki@yahoo.gr (E.V.); 3Department of Internal Medicine, General Hospital of Katerini, 60100 Katerini, Greece

**Keywords:** adrenal insufficiency, β-thalassemia, diabetes mellitus, endocrinopathy, hypothyroidism, hyperparathyroidism, hypogonadism, hemoglobinopathies, osteoporosis, sickle cell disease

## Abstract

Hemoglobinopathies, including β-thalassemia and sickle cell disease (SCD), are common genetic blood disorders. Endocrine disorders are frequent manifestations of organ damage observed mainly in patients with β-thalassemia and rarely in SCD. Iron overload, oxidative stress-induced cellular damage, chronic anemia, and HCV infection contribute to the development of endocrinopathies in β-thalassemia. The above factors, combined with vaso-occlusive events and microcirculation defects, are crucial for endocrine dysfunction in SCD patients. These endocrinopathies include diabetes mellitus, hypothyroidism, parathyroid dysfunction, gonadal and growth failure, osteoporosis, and adrenal insufficiency, affecting the quality of life of these patients. Thus, we aim to provide current knowledge and data about the epidemiology, pathogenesis, diagnosis, and management of endocrine disorders in β-thalassemia and SCD. We conducted a comprehensive review of the literature and examined the available data, mostly using the PubMed and Medline search engines for original articles. In the era of precision medicine, more studies investigating the potential role of genetic modifiers in the development of endocrinopathies in hemoglobinopathies are essential.

## 1. Introduction

Hemoglobinopathies include a group of genetic syndromes, with β-thalassemia and sickle cell disease (SCD) being the most common [1]. β-thalassemia is an inherited hematological disorder characterized by reduced production or complete absence of beta-globin chains, a component of adult hemoglobin (HbA) [2]. Three types of β-thalassemia have been described: β-thalassemia major, in which regular life-long transfusions are required, intermedia, and minor [3]. Furthermore, depending on transfusion requirements, β-thalassemia is classified as transfusion-dependent (TDT) or non-transfusion-dependent (NTDT) [3]. Patients with this condition are prone to developing various endocrine disorders, with chronic anemia, hypoxia, and iron overload having a pathophysiological role [2,4].

Sickle cell disease (SCD) is an inherited hemoglobinopathy characterized by the presence of Hemoglobin S and caused by a single gene mutation [5]. SCD patients suffer from hemolytic anemia, vaso-occlusive events of small vessels, and organ dysfunction [5]. Recently, many advances have been made in understanding the pathophysiology of SCD and its complications, with complement system activation having a leading role in those [6,7,8]. Frequent transfusions are among the most common therapies used to treat anemia in SCD, leading to iron overload, which results in oxidative stress-mediated organ damage [9]. Thus, endocrine dysfunction is prevalent in SCD patients due to iron overload and micro-circulation defects. In this critical review, we intend to provide an up-to-date summary of knowledge about endocrinopathies affecting patients with hemoglobinopathies (β-thalassemia and SCD). Our primary aim is to describe the pathogenetic mechanisms implicated in the development of endocrine disorders in individuals with hemoglobinopathies, while our secondary aims are to review the prevalence, the risk factors, the diagnostic methods, and the treatment options of endocrine diseases in β-thalassemia and SCD.

## 2. Methods

A comprehensive review of the literature was conducted, and all the available data were critically examined using the PubMed and Medline search engines for original articles. Full-text articles published in English, based on our search, were included. The keywords of research were “β-thalassemia” or “sickle cell” in combination with one of the following: “endocrine”, “glycemia”, “diabetes”, “thyroid”, “TSH”, “parathyroid”, “parathormone”, “osteoporosis”, “bone mineral density”, “hypogonadism”, “testosterone”, “growth”, “GH”, “adrenal” or “ACTH”. Clinical studies (cross-sectional studies, retrospective analyses, cohort studies, clinical trials, case–control studies) in both pediatric and adult populations with β-thalassemia or SCD, meta-analyses, and clinical practice guidelines were included in this review, aiming mainly to clarify the prevalence of endocrinopathies in populations affected by hemoglobinopathies and the pathogenetic mechanisms implicated in their development.

## 3. Impaired Glucose Metabolism and Diabetes Mellitus

Impaired glucose metabolism in β-thalassemia implicates increased insulin resistance, glucose intolerance, and diabetes mellitus (DM) [4]. In Table 1, the prevalence of DM in β-thalassemia is reported and varies from 0% to 35% [10,11,12,13,14,15]. Risk factors for impaired glucose metabolism include older age, high body mass index (BMI), elevated mean ferritin serum levels, and history of splenectomy [11,12,16,17,18]. Impaired insulin secretion due to iron deposition in the pancreas, insulin resistance, iron-induced fatty acid oxidation with a subsequent decrease in glucagon usage rate, and zinc deficiency contribute to the pathogenesis of DM in β-thalassemia [4,19,20,21,22].

Furthermore, viral hepatitis (HCV infection), chronic liver disease, and cirrhosis are considered to be among the major causes of impaired glucose metabolism in TDT [4,11,17,20,21,23,24]. Interestingly, cytokines such as tumor necrosis factor-a (TNF-a) may contribute to peripheral insulin resistance [19]. The development of diabetes mellitus has been correlated with the presence of IVS-II nt 745 genotype [21,23].

In accordance with international guidelines, fasting glucose levels should be evaluated twice a year, starting at the age of 5 years old, and an oral glucose tolerance test (OGTT) should be performed if the level exceeds 110 mg/dL. In addition, a 2 h 75 g oral glucose tolerance test (2 h OGTT) should be performed at the ages of 10, 12, 14, and 16 years old, plus every year thereafter [4,22]. In Table 2, the diagnostic criteria of the Thalassemia International Federation (TIF) for the diagnosis of DM and impaired glucose tolerance (IGT) are presented [25]. Moreover, even in the absence of abnormal fasting and 2 h OGTT glucose levels, hyperglycemia can be detected by a continuous glucose monitoring system (CGMS) [22]. HbA1c is not considered a reliable biomarker for long-term evaluation of glucose control in TDT [4]. At present, there is not a single laboratory marker that can safely reflect long-term glucose control in patients with TDT. However, glycated albumin and fructosamine levels have been used to predict glycemic control in patients with β-thalassemia [22,26,27]. A Pancreas T2* Magnetic Resonance Imaging (MRI) might reduce the need for OGTT and detect high-risk patients [22,28]. It has been suggested that pancreatic iron concertation could be an indicator of beta-cell toxicity and impaired glucose metabolism [28,29,30]. On the contrary, Hashemieh et al. did not find significant differences between diabetic and non-diabetic thalassemic patients regarding the MRI T2* or the status of pancreatic iron overload [23].

Among the most significant measures for preventing and treating glucose homeostasis disturbances in patients with β-thalassemia are iron chelation therapy, prevention, and management of chronic HCV infection. Patients, especially in the early stages, benefit from iron chelation therapy with deferoxamine plus deferiprone [4]. Combining these iron-chelating agents reduced insulin resistance and increased ß-cell function (as measured by relevant indices) in comparison to patients who received monotherapy and had opposite results [31]. Similarly, in the study of Farmaki et al., co-administration of these agents was correlated with lower mean serum ferritin levels, decreased hepatic iron deposition, and decreased glucose intolerance [32]. Initial management of hyperglycemia in these patients includes lifestyle modification combined with an oral glucose-lowering agent [33]. De Sanctis et al. showed in their retrospective study that metformin was prescribed to approximately half of the patients with β-thalassemia and DM (47.6%) [34]. The majority of patients who received oral glucose-lowering agents (61.2%) were treated with just a single agent [34]. In the case of overt DM and providing that all other measures have failed, patients with β-thalassemia require treatment with subcutaneous insulin injections (basal, bolus, or combination) [4,33]. It would be interesting to investigate the effect of dapagliflozin in patients with β-thalassemia, an SGLT2 inhibitor, as it was found that it contributes to anemia improvement in non-thalassemic patients with DM type 2 [35].

A lower prevalence of DM in SCD individuals compared to the general population has been reported in several studies (Table 3) [36,37,38,39,40,41]. Increased risk for DM is associated with older age and higher BMI [36]. In a cohort of 7070 SCD African American patients, the standardized prevalence of T2D was estimated from 15.7% to 16.5% [42]. The lower prevalence of DM in SCD individuals can be attributed to decreased rates of obesity and lower life expectancy in comparison to the general population [37,41,43]. However, given the advances made in the management of SCD, leading to an increase in the life expectancy of those patients, it is reported that the prevalence of DM will increase in SCD populations [42,44]. Diabetic complications such as retinopathy and impaired renal function are more prevalent in patients with SCD and T2D than in those with T2D only [45]. In the same study, the expression of E-selectin, a plasma biochemical marker of endothelial dysfunction expressed in human aortic endothelial cells, was higher in diabetic SCD subjects. Diabetic complications (proteinuria and retinopathy) might be more prevalent in male-gender SCD patients [46].

Interestingly, it has been found that normoglycemic SCD patients have impaired pancreatic β-cell function and diminished insulin production [47]. Fung et al. showed that iron overload, due to transfusion therapy, plays a substantial role in the pathogenesis of T2D in those patients [43]. In patients with sickle cell disease and DM, HbA1c is not reliable for the diagnosis of DM and monitoring of glycemic control, while serum glucose measurements should be used [48]. Furthermore, the levels of serum fructosamine have been used for the monitoring of those patients [49,50,51]. Metformin might reduce the occurrence of hospitalization rates in SCD individuals and vaso-occlusive episodes [52]. This could be attributed to an increase in fetal hemoglobin (HbF) production, as suggested by laboratory data [53]. Smiley et al. suggested that in the majority of SCD patients with T2D, a combination of oral agents and insulin therapy is required for successful glycemic control [9]. Given the fact that SCD-overweight individuals face a higher risk for T2D, dietary intervention and lifestyle modification are essential.

**Table 1 ijms-24-16263-t001:** Prevalence, risk factors, and pathogenetic mechanisms of endocrinopathies in β-thalassemia.

Endocrine Disorder	Prevalence	References	Risk Factors–Pathogenesis
Diabetes mellitus	0–35%	[10,11,12,13,14,54]	Older age, higher BMI, splenectomy, iron overload, HCV infection, chronic liver disease, zinc deficiency
Hypothyroidism	6.2–52%	[15,55,56,57,58,59,60,61,62,63]	Older age, splenectomy, chronic anemia, iron overload ^1^
Hypoparathyroidism	1–38%	[15,61,64,65,66]	Iron accumulation in parathyroid glands, increased serum ferritin levels, chronic anemia, magnesium deficiency
Osteoporosis	12.5–62%	[67]	Vitamin D deficiency, iron overload, bone marrow hyperplasia, endocrine disorders
Hypogonadism	20–77.5%	[68,69,70,71]	Iron accumulation in the pituitary gland, liver, ovaries, and testes; chronic anemia; HCV infection
Growth hormone deficiency	26.6%	[72]	Increased serum ferritin levels, hepatic iron overload
Adrenal insufficiency	13–46%	[73,74]	Iron overload, chronic anemia

BMI, body mass index; HCV, hepatitis C virus. ^1^ In some studies, iron overload, as measured by serum ferritin, is not correlated with thyroid gland dysfunction.

**Table 2 ijms-24-16263-t002:** Diagnostic criteria of Thalassemia International Federation for the diagnosis of diabetes mellitus and impaired glucose tolerance in β-thalassemia [25].

Glucose Metabolism Disorder	Criteria for the Diagnosis
DM	Fasting serum glucose > 126 mg/dL
	Serum glucose at 2 h > 200 mg/dL in 2 h OGTT
IGT	Serum glucose at 2 h > 140 mg/dL in 2 h OGTT

DM, diabetes mellitus; IGT, impaired glucose tolerance; OGTT, oral glucose tolerance test.

**Table 3 ijms-24-16263-t003:** Prevalence, risk factors, and pathogenetic mechanisms of endocrinopathies in sickle cell disease.

Endocrine Disorder	Prevalence	References	Risk Factors–Pathogenesis
Diabetes mellitus	1.46–19%	[36,37,38,39,40,41,42]	Older age, higher BMI, iron overload
Hypothyroidism	0–18.1%	[75,76,77,78,79,80,81]	Hepatic iron overload, thyroid microcirculation dysfunction, iron overload1 ^1^
Secondary hyperparathyroidism	8.7–71.4%	[82,83,84,85]	Vitamin D deficiency
Osteoporosis	7–40%	[86,87,88,89,90,91]	Calcium and vitamin D deficiency, bone vaso-occlusion-induced ischemia
Hypogonadism	24% ^2^	[92]	Vaso-occlusive events in vessels of hypothalamic–pituitary system, iron overload
Adrenal insufficiency	19.4% ^3^	[93]	Vaso-occlusive events in vessels of hypothalamic–pituitary system and adrenal glands, iron overload, chronic use of opioid analgesics

BMI, body mass index. ^1^ In some studies, iron overload, as assessed by serum ferritin, is not associated with thyroid dysfunction. ^2^—as suggested in a study of 34 male patients. ^3^—as found in a cohort of 62 hospitalized sickle cell disease patients with painful crises.

## 4. Hypothyroidism

Thyroid dysfunction in TDT includes primary, subclinical, and secondary hypothyroidism. The prevalence of hypothyroidism is presented in Table 1 and ranges from 6.2 to 52% [15,55,56,57,58,59,60,61,62,63]. Subclinical hypothyroidism was found in 6.7–43.3% of patients with major or intermedia β-thalassemia [36,39,40,42,43,44]. The frequency of clinical hypothyroidism also varies from 0 to 18.3% [55,56,57,58,59]. According to Abdel-Razek et al., there is no statistically significant difference between patients with thalassemia major and those with thalassemia intermedia as regards the prevalence of hypothyroidism [56]. In thalassemia intermedia, multiple factors are considered to be involved in the pathogenesis of hypothyroidism, including chronic hypoxia and anemia [56]. Taher et al. reported a statistically significant association between thyroid dysfunction with age and a history of splenectomy in patients with thalassemia intermedia [94]. Regular blood transfusions may lead to iron accumulation in the thyroid gland with subsequent glandular parenchyma fibrosis [56,62,95]. However, iron overload, as measured by serum ferritin, is not associated with thyroid dysfunction in several studies, despite claims to the contrary [55,56,61,62,63,96,97]. Chronic hypoxia, oxidative stress, and mitochondrial and lysosomal damage, as well as subsequent cellular death, contribute to thyroid impairment in thalassemic patients [56,59,95]. All patients with overt or subclinical hypothyroidism should begin treatment as soon as their TSH level exceeds 10 mIU/L. Moreover, in patients with TSH levels below or equal to 10 mIU/L, with co-existing infertility, goiter, or positive anti-thyroid antibodies, initiation of treatment is recommended [98]. Abdel-Razek et al. reported that, at their center, subclinical hypothyroidism was not treated and that hormonal replacement was exclusively administered to patients with overt hypothyroidism [56].

Abnormalities in thyroid gland function, including subclinical (the most frequent) primary and secondary hypothyroidism, have been reported in SCD individuals [99,100,101]. The prevalence of hypothyroidism is estimated from 0 to 18.1%, as shown in Table 3 [75,76,77,78,79,80,81]. Even in SCD individuals with normal clinical and laboratory thyroid function, TSH levels have been found to be higher compared to normal controls [76,102,103]. Subclinical hypothyroidism in children has been associated with male gender and the presence of constipation [79]. Yassin et al. reported that hypothyroidism is more prevalent in SCD patients with hepatic iron overload [104]. Various studies have shown a diminished thyroid gland volume in SCD subjects compared to healthy controls, and this has been attributed to thyroid microcirculation dysfunction [75,76]. Thyroid dysfunction has not been associated with iron overload and high levels of serum ferritin in some studies [75,102]. However, Garadah et al. reported that there is a negative correlation between serum ferritin and fT4 levels [77]. Iron deposition to the thyroid gland might cause cellular damage and provoke primary thyroid failure [9]. In Figure 1, we summarize the main pathogenetic mechanisms that are implicated in the development of thyroid disorder in SCD and β-thalassemia.

## 5. Parathyroid Dysfunction

Hypoparathyroidism (HPT) occurs in a range of 1–38% in patients with β-thalassemia major (Table 1) [15,61,64,65,66]. HPT was diagnosed in 4.4% of patients with thalassemia intermedia [64]. In thalassemic patients, especially in those with HPT, median plasma Fibroblast Growth Factor 23 (FGF-23) levels were significantly lower compared to healthy controls [52]. However, in both groups, PTH levels and 25-hydroxyvitamin-D levels did not differ significantly [66]. Iron accumulation in parathyroids and insufficient treatment with iron chelation agents are associated with the development of HPT in β-thalassemia patients [64]. Based on data collected from a group of patients with β-thalassemia major, Saki et al. found a strong correlation between 1,25(OH)2D3, FGF23, and ferritin levels [105]. In addition, in the study of De Sanctis et al., approximately half of the patients (48.9%) with β-thalassemia (major or intermedia) and hypoparathyroidism had a serum ferritin level over 2500 ng/mL [64]. Other factors contributing to parathyroid dysfunction include chronic anemia and magnesium deficiency [64]. It was found that the occurrence of HPT was significantly associated with age, the mean number of transfusions received, the total number of transfusions per year, splenomegaly, the history of splenectomy, hepatomegaly, and type of chelation therapy [61]. The majority of β-thalassemia patients with HPT are asymptomatic [106]. However, in the study of De Sanctis et al., 14.2% of the patients suffered from life-threatening symptoms, such as spasms of the larynx, arrhythmias, and cardiac failure [64]. Karimi et al. reported that 54.2% of β-thalassemia patients with HPT had intracerebral calcifications, most frequently in the frontoparietal cortex and basal ganglia [107]. Intracranial calcium depositions should be documented with MRI in every case of symptomatic hypoparathyroidism [106].

Levels of parathyroid hormone have been reported to be higher in SCD patients in comparison to healthy individuals [108]. Secondary hyperparathyroidism affects 8.7–71.4% of SCD patients, as shown in various studies (Table 3) [82,83,84,85]. Vitamin D deficiency contributes to the development of secondary hyperparathyroidism in these patients (Figure 1) [109]. Furthermore, primary hyperparathyroidism can affect mainly young subjects with SCD and has been correlated with a high risk of low bone mineral density (BMD) [110,111,112].

## 6. Bone Mineral Density

A reduction in BMD in β-thalassemia patients can be caused by both genetic and environmental factors, which either restrain the activity of osteoblasts or induce osteoclast activity [67]. Osteoporosis in β-thalassemia is most prominent among male patients and can be attributed to many factors, such as endocrine disorders (parathyroid gland dysfunction, DM, hypothyroidism, hypogonadism), impairment in GH-IGF-I axis (growth hormone–insulin-like growth factor I), vitamin D deficiency, iron overload, bone marrow hyperplasia, liver disease, and cardiomyopathy (Table 1) [67,106]. According to TIF recommendations, evaluation of bone health in individuals with β-thalassemia should be initiated by the age of ten years, while the values of serum and urine calcium and phosphate, as well as serum alkaline phosphatase, vitamin D, and PTH should be monitored yearly [25]. Dual-energy X-ray absorptiometry (DXA) is used once a year from the age of ten years old to evaluate BMD and estimate the risk of vertebral fracture. Treatment options include administration of vitamin D and calcium supplements or hormonal replacement. In the case of established osteoporosis, bisphosphonates are provided, while the use of novel agents of osteoclast inhibition (denosumab, teriparatide, sotatercept) is recommended, although their effectiveness in patients with β-thalassemia has not been proven yet [25].

Mineral bone disorders are common in children and adult patients with SCD. Patients with SCD have lower BMD in comparison to healthy subjects [86,87]. The prevalence of osteoporosis ranges from 7% to 40%, as reported in several studies [86,87,88,89,90,91]. In a cross-sectional study, it was found that 72% of SCD patients had low BMD and that the lumbar spine is the most prevalent anatomic site affected [90]. Risk factors and mechanisms for diminished BMD in SCD include calcium and vitamin D deficiency, delayed puberty development, and bone vaso-occlusion-induced ischemia [9]. Interestingly, low BMD in SCD has been associated with low serum magnesium [110]. Franceschi et al. recently developed an algorithm for the clinical management of bone disease in SCD (SBD) and showed that a spine radiograph is superior to BMD evaluation for the diagnosis and follow-up of (SBD) [91].

## 7. Gonadal Failure

Hypogonadism, especially hypogonadotropic, is considered to be the most prevalent endocrine complication in β-thalassemia, with a prevalence from 20 to 77.5%, affecting both adults and adolescents (Table 1) [56,68,69,113]. Hypogonadism has been correlated with the age and genotype of patients with β-thalassemia, average Hb levels, serum ferritin levels, and co-existence with DM, HCV infection, and liver disease [69,70,71,114]. Iron accumulation in the pituitary gland, chronic anemia, and increased oxidative stress might play a significant role in the development of secondary gonadal failure in those patients [70,114,115,116]. Gonadotropic cells appear to be the most susceptible to iron cytotoxicity [68,113]. Primary hypogonadism is a result of iron deposition and irreversible damage of the ovaries and testis of patients with β-thalassemia [114,117]. Interestingly, iron deposition on adipose tissue may also contribute to hypogonadotropic hypogonadism in those patients [70]. Moreover, iron accumulation in the endometrium can affect fetus implantation, explaining the link between infertility and iron overload [118]. Bronspiegel et al. showed that the administration of deferoxamine in young patients (under ten years old) might prevent gonadal failure and delayed puberty [119]. Impaired hepatic synthesis of IGF-1 due to deficient growth hormone (GH) secretion, hepatic iron overload, and inadequate nutrition may contribute to hypogonadism in patients with β-thalassemia [114].

As a result of hypogonadism, 40–80% of patients with TDT suffer from sexual dysfunction, pubertal delay, deficient sexual development, and osteoporosis [69,114,116,118]. The diagnostic approach for the diagnosis of hypogonadism in β-thalassemia patients should include evaluation of FSH, LH, sex steroids, and prolactin serum levels [114,118]. In cases of ambiguous results, the gonadotropin-releasing hormone analog stimulation test should be performed [116]. Moreover, R2* MRI of the pituitary gland seems to be a useful tool to estimate the gland’s size and detect patients with β-thalassemia and secondary hypogonadism due to iron deposition on the pituitary gland [68,120]. Serum ferritin levels can be used as a predictor of sexual development in patients with β-thalassemia [17]. Low doses of sex steroids (ethinyl estradiol, intramuscular testosterone esters) are used for the treatment of gonadal failure [25,114]. The use of combined human chorionic gonadotropin and recombinant follicle-stimulating hormone has been found promising for pubertal induction [70,121]. Another potential treatment option for hypothalamic hypogonadism is pulsatile gonadotrophin-releasing hormone therapy [121]. When testosterone replacement therapy is introduced in patients with hypogonadism-related osteoporosis, bisphosphonates should be co-administered [114]. In females, a combination of estrogens with progesterone is commonly used [118]. Furthermore, assisted reproductive technology treatments, specifically in vitro fertilization, might be beneficial to infertile individuals with β-thalassemia [122].

Hypogonadism has been suggested to be one of the most common endocrinopathies in SCD (Table 3) [9]. Testosterone serum levels have been found to be diminished in male SCD patients compared to healthy subjects, as well as the prevalence of hypogonadism 24% in a cohort of 34 patients [92,100,123,124,125]. Some studies indicate a central etiology of hypogonadism in SCD (as suggested by decreased gonadotropin serum levels), while others reported that most cases are caused by primary testicular failure [92,100,123,124,125,126,127]. Ribeiro et al. reported in their cohort a high prevalence of compensated hypogonadism, characterized by increased levels of FSH and LH and normal levels of testosterone [128]. Furthermore, Martins et al. proposed androgen resistance as a possible mechanism of hypogonadism in SCD (peripheral hypogonadism) [129]. Primary hypogonadism has been suggested as the major cause of abnormal testosterone levels, attributed to increased oxidative stress, leading to diminished transport of cholesterol to mitochondria of Leydig cells [130,131]. Recurrent testicular infractions could contribute to the development of primary gonadal failure in these men [132,133]. Secondary testosterone deficiency affects male individuals with a more advanced form of SCD [100]. Vaso-occlusive events in small vessels of the hypothalamic–pituitary system and transfusion iron overload might play a significant role in the development of secondary hypogonadism [134,135]. Eunuchoid habitus, small testicular size, absence of secondary sexual characteristics, poor libido, and infertility are manifestations of hypogonadism in SCD male individuals [9,123]. Reduced testosterone levels have been associated with priapism in SCD males [130]. Testosterone replacement therapy and clomiphene have been administrated to mitigate the symptoms of hypogonadism in SCD male subjects with variable outcomes [136,137]. Infertility is more prevalent in men with SCD compared to female SCD individuals [138]. Yet, hypogonadotropic hypogonadism also affects women with SCD [139].

## 8. Growth Failure

Children with TDT are often diagnosed with growth failure [113]. In a meta-analysis of seventy-four studies, the pooled prevalence of short stature, growth failure, and growth hormone deficiency was reported to be 48.9%, 41.1%, and 26.6%, respectively, higher in the male population with β-thalassemia [72]. Singh et al. found that final stature in children with TDT was significantly correlated with the arrest of puberty, while all adolescents receiving hormone replacement therapy had decreased height [116]. Additionally, growth retardation has been correlated with elevated serum ferritin, while short stature has been associated with serum ferritin levels and hepatic iron overload, as expressed by liver iron concentration (LIC) [104,114]. As opposed to this, Grundy et al. found no significant association between short stature and iron chelation therapy, suggesting that other factors (socioeconomic, racial, genetic) are implicated [115]. Growth retardation and body disproportion, caused by flattened vertebral bodies, have been considered as an outcome of various factors, such as chronic anemia and hypoxia, an increase in basal metabolic rate (BMR), iron concentration in the endocrine glands as well as in erythrocytes, deferoxamine’s use, zinc and folic acid deficiency and co-existing gonadal failure, DM, and liver disease [72].

Growth failure is frequent in SCD populations [9,140]. Young SCD individuals have significantly lower stature and BMI compared to healthy subjects, as has been shown in several studies [141,142,143]. Endocrine abnormalities, iron overload, decreased appetite, nutritional deficiencies, increased BMR, and gonadal failure contribute to the pathogenesis of delayed growth in SCD [9,104,144]. Growth failure has been associated with abnormalities in the GH-IGF-I axis (growth hormone–insulin-like growth factor I) [145,146]. Serum levels of IGF-I have been reported to be diminished in children with SCD, attributed to malnutrition and defects in the GH-IGF-I axis [104,140,145,146]. Defects in GH-IGF-I and hypothalamic–pituitary axis might be a result of vaso-occlusive events during sickle cell crisis episodes [147].

## 9. Adrenal Insufficiency

The prevalence of adrenal insufficiency (AI) ranges between 13 and 46% in patients with β-thalassemia (Table 1) [39,73,74]. It is reported that AI is more common in male patients with β-thalassemia [73]. Portadown et al. found that thalassemia patients affected by AI remain asymptomatic and rarely present clinical symptoms [74]. Iron overload and oxidative stress, chronic anemia, hypoxia, and the direct effect of chelation agents are implicated in the pathogenesis of adrenal dysfunction in those patients and might cause dysregulations in the hypothalamic–pituitary–adrenal axis [73]. According to TIF guidelines, an assessment of adrenal function should be performed every 1–2 years, particularly in the case of co-existing GH deficiency. As most cases remain subclinical, treating adrenal insufficiency with glucocorticoids is only necessary in highly stressful situations [25].

Adrenal dysfunction also affects patients suffering from SCD. Plasma cortisol levels had been found diminished in both adult and pediatric patients suffering from SCD compared to healthy controls, while in children with SCD (in steady state) and SS genotype were significantly lower compared to children with AA and AS genotypes [100,139,148]. Crisis-free SCD patients have been found to have relatively defective adrenal function compared to healthy controls [149]. Rosenbloom et al. showed that plasma cortisol levels were decreased during painful crises as assessed by the insulin hypoglycemia test [150]. Interestingly, Saad et al. proposed a central origin for the decreased cortisol production in SCD. Specifically, in this study, the response to the ACTH stimulation test had no significant differences between SCD patients during the steady phase and normal controls [151]. The prevalence of adrenal insufficiency, assessed by cosyntropin testing, was found to be 19.4% in hospitalized SCD patients with painful crises, and the origin of AI was secondary to pituitary or hypothalamic disease in all cases [93]. Recently, Hollister et al. indicated that hair cortisol levels were decreased in SCD individuals with a more severe form of the disease [152]. Oxidative stress due to iron overload syndrome, secondary to multiple red blood cell transfusions, contributes to adrenal gland dysfunction in patients suffering from SCD [9,43,93,153]. Furthermore, adrenal insufficiency in SCD has been attributed to both hemorrhage and thromboembolic events causing recurrent occlusion of small vessels, while vaso-occlusion-induced ischemia might involve the brain, pituitary gland and the adrenal glands in those patients [9,154,155]. Chronic use of opioid analgesics is common for the management of pain in SCD and might be a potential cause of secondary AI [93,156].

## 10. Genetic Modifiers of Endocrinopathies in Hemoglobinopathies

Decreased expression of apolipoprotein B in peripheral blood monocytes, due to the presence of the rs59014890 single nucleotide polymorphism (SNP) in the C allele of the gene, was correlated with higher DM risk in those affected by SCD [36]. Candidate gene association studies have been widely used in order to find the genes implicated in the pathogenesis of osteoporosis in β-thalassemia. Rs4701 variant in the VDBP gene, which encodes vitamin D binding protein, has been associated with lower BMD in β-thalassemia patients, as shown in a case–control study [157]. The presence of a polymorphism at the Sp1 binding site of the collagen type I A1 (COLIA1) gene (G > T, at first intron) is considered to be a major factor in the development of low BMD and osteoporosis in both patients with β-thalassemia and in the general population [158,159,160]. FokI and BsmI polymorphisms at the Vitamin D receptor (VDR) gene have been correlated with low BMD and increased risk of osteoporosis in several studies [161,162,163,164,165]. Those data are presented in Table 4. Understanding the role of underlying genetic modifiers in the pathogenesis of endocrine disorders in hemoglobinopathies is crucial in the era of precision medicine to improve patients’ outcomes. The International Hemoglobinopathy Research Network (INHERENT) works in this direction, studying the genetic modifiers of hemoglobinopathies with a large-scale, multi-ethnic genome-wide association study (GWAS) [166].

## 11. Conclusions

Regular blood transfusions, adequate chelation therapy, and novel therapies have improved the life expectancy and life quality of patients with hemoglobinopathies. The pathogenesis of endocrine dysfunction in β-thalassemia is multifactorial and involves iron overload, chronic anemia, and hypoxia. In SCD, vaso-occlusive events and defects in microcirculation are also implicated. Furthermore, endocrine disorders could affect children, adolescents, and adult patients with β-thalassemia and SCD. Older age has been identified as a risk factor for the development of several endocrinopathies in some studies. However, more studies focusing on the differences in the prevalence of endocrine disorders between the various age groups are essential. Physicians should evaluate young patients with β-thalassemia for endocrinopathies to diagnose them promptly. Much effort must be made to improve patients’ access to chelation therapy, especially in countries of the developing world. Furthermore, the development of guidelines on the diagnostic work-up and management of endocrine disorders in SCD individuals is crucial. In the era of precision medicine, GWASs are vital to identify potential genetic modifiers of endocrinopathies in hemoglobinopathies. Thus, novel treatment plans could be developed, contributing to an improved quality of life in these patients.

## Figures and Tables

**Figure 1 ijms-24-16263-f001:**
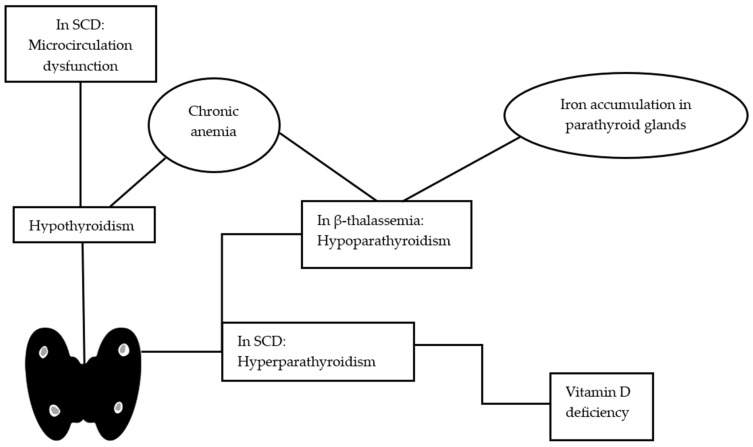
Overview of the pathophysiological insights in the pathogenesis of thyroid and parathyroid disorders in hemoglobinopathies. SCD, sickle cell disease.

**Table 4 ijms-24-16263-t004:** Studies investigating the potential role of genetic modifiers in the pathogenesis of endocrinopathies in hemoglobinopathies.

First Author Publication Year Reference Number	Study Type	Disease	Gene	Trait
Zhang 2015 [36]	Meta-analysis of cohorts	SCD	C allele of rs59014890 SNP	DM
Sahmoud 2020 [157]	Case–control	β-thalassemia	rs4701 SNP at the VDBP	Osteoporosis
Perrotta 2000 [158]	Case–control	β-thalassemia	Sp1 binding site SNP at the COLIA1 gene	Osteoporosis
Arisal 2002 [159]	Cohort	β-thalassemia	Sp1 binding site SNP at the COLIA1 gene	Low BMD
Hamed 2011 [160]	Case–control	β-thalassemia	Sp1 binding site SNP at the COLIA1 gene	Osteoporosis
Ragab 2016 [165]	Case–control	β-thalassemia	Ile105Val polymorphism at GSTP1 gene	Low BMD
Singh 2012 [164]	Cohort	β-thalassemia	FokI and BsmI polymorphisms at VDR gene	Low BMD
Abbassy 2019 [163]	Case–control	β-thalassemia	FokI and BsmI polymorphisms at VDR gene	Osteoporosis
El-Edel 2010 [162]	Cohort	β-thalassemia	BsmI polymorphisms at VDR gene	Osteoporosis
Ferrara 2002 [161]	Cohort	β-thalassemia	FokI and BsmI polymorphisms at VDR gene	Low BMD Growth failure

SCD; sickle cell disease, SNP; single nucleotide polymorphism, DM; diabetes mellitus, VDBP; vitamin D binding protein, Sp1; specificity protein 1, COLIA1; collagen type I alpha 1, GSTP1; glutathione S-transferase pi 1, BMD; bone mineral density, VDR; vitamin D receptor.

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
