# Peer review of "Endocrinopathies in Hemoglobinopathies: What Is the Role of Iron?"

_ijms, 2023, doi:10.3390/ijms242216263_

Round 1

Reviewer 1 Report

Comments and Suggestions for Authors

Congratulations to the authors for a comphrensive review. It was a pleasure to read. 

Please find some suggestions that would hopefully benefit the manuscript.

Language:

The language is of a high standard without a need for language revision.

The title:

The reviewer likes the title, but sadly it is misleading. When we link this with the aim, it is as if the authors already decided the answer making the title irrelevant. The scope of the aim “In this review we intend to provide an up-to-date summary of knowledge about endocrinopathies affecting patients with hemoglobinopathies (β-thalassemia and SCD), focusing on the prevalence, pathogenetic mechanisms, risk factors, diagnostic methods, and treatment options of endocrine diseases in patients with β-thalassemia or SCD.” encompasses far more than the title suggests.

-          Is it possible for the authors to compose a primary and secondary aim to their manuscript? i.e. for the factors contributing to endocrinopathies to be the primary aim and the prevalence, pathogenesis and management to be secondary aims?

-          The authors have structured the manuscript this manner anyway.

A more appropriate tile (but not perfect) would then be: Endocrinopathies in Hemoglobanopathies: what is the role of Iron?

Methodology:

-          The authors have not clearly defined the type of review they did.

Types of Reviews - Systematic Reviews - LibGuides at Duke University Medical Center

The reviewer assumes it is a critical review, but it isn’t clear from L 57-58.

-          It is important for the authors to be clear regarding the type of review that readers may also understand the bias in the manuscripts reviewed and the level of evidence that is presented in this manuscript.

-          If this screening process has not been done then it either has to be clear from the type of review done, or the authors must declare the limitations of the review.

Conclusion:

-          The statement of the authors L 427 to 429 “Much effort must be made to improve patients’ compliance with chelation therapy, especially in countries of the developing world.” Is inappropriate and simplified. The problem with chelation therapy compliance in lower resourced settings are matters of finances, availability of medication, access to medication etc. Factors that are not in the power of patients to improve.

Furthermore, the authors have not argued the point of compliance in their review, but rather efficacy.

The reviewer suggests that the authors rethink this statement and rewrite it in line with the content of the manuscript.  

Tables:

Table 1

-          The risk factors from one endocrine disorder flows into another, making it hard to distinguish the one row from another.

-          Equally the column of references are not in line with each row. This is a table that needs clearer delineation between column and rows

Table 2 and 3 suffers from the same problem

Table 4 reads well but the alignment of the last publication needs attention.

Figure 1:

-          White writing on a light coloured background reads very poorly. The reviewer suggests that the writing should be in black please.

-          The compilation of the figure is not comprehensible. Is there a different meaning between a circle and a rectangle?

-          Why are some lines connecting to some circles and not others?

In totality the figure is confusing. May the authors reconstruct the figure in a manner that the message is clear?

General:

-          One question that the reviewer had hope would be answered is the difference in endocrinopathies between children and adolescent populations versus adults. Did the authors find any answers in the literature?

-          And when in a patient’s lifespan these endocrinopathies are most prevalent to happen?

Author Response

Authors' response

Congratulations to the authors for a comprehensive review. It was a pleasure to read. 

Answer: Dear Reviewer, we would like to thank you for the time that you dedicated to reading and reviewing our work. Your comments and suggestions were valuable for us and helpful for improving our review.

Please find some suggestions that would hopefully benefit the manuscript.

Language:

The language is of a high standard without a need for language revision.

Answer: We appreciate this comment.

The title:

The reviewer likes the title, but sadly it is misleading. When we link this with the aim, it is as if the authors already decided the answer making the title irrelevant. The scope of the aim “In this review we intend to provide an up-to-date summary of knowledge about endocrinopathies affecting patients with hemoglobinopathies (β-thalassemia and SCD), focusing on the prevalence, pathogenetic mechanisms, risk factors, diagnostic methods, and treatment options of endocrine diseases in patients with β-thalassemia or SCD.” encompasses far more than the title suggests.

-          Is it possible for the authors to compose a primary and secondary aim to their manuscript? i.e., for the factors contributing to endocrinopathies to be the primary aim and the prevalence, pathogenesis and management to be secondary aims?

      Answer: We thank the reviewer for this valuable idea. The primary and secondary aims of the review are now included in the manuscript, as you suggested (Lines 55-59).

-          The authors have structured the manuscript this manner anyway.

A more appropriate tile (but not perfect) would then be: Endocrinopathies in Hemoglobinopathies: what is the role of Iron?

Answer:  We are grateful for the reviewer’s comments and suggestions. We changed the title to the one you suggested.

Methodology:

-          The authors have not clearly defined the type of review they did.

Types of Reviews - Systematic Reviews - LibGuides at Duke University Medical Center

The reviewer assumes it is a critical review, but it isn’t clear from L 57-58.

-          It is important for the authors to be clear regarding the type of review that readers may also understand the bias in the manuscripts reviewed and the level of evidence that is presented in this manuscript.

-          If this screening process has not been done then it either has to be clear from the type of review done, or the authors must declare the limitations of the review.

       Answer: We are thankful for these comments. The reviewer is right. Indeed, we did not mention that our work was a critical review. We included this in the introduction section (Line 53).

Conclusion:

-          The statement of the authors L 427 to 429 “Much effort must be made to improve patients’ compliance with chelation therapy, especially in countries of the developing world.” Is inappropriate and simplified. The problem with chelation therapy compliance in lower resourced settings are matters of finances, availability of medication, access to medication etc. Factors that are not in the power of patients to improve.

Furthermore, the authors have not argued the point of compliance in their review, but rather efficacy.

The reviewer suggests that the authors rethink this statement and rewrite it in line with the content of the manuscript.  

Answer: We are grateful for this comment. We agree with the suggestion of the reviewer. Indeed, in developing countries, compliance is not the problem. In those countries, the limited access to adequate chelation therapy constitutes a major issue. We rephrased our conclusion, as you suggested (Lines 447-448). Thanks once again.

Tables:

Table 1

-          The risk factors from one endocrine disorder flows into another, making it hard to distinguish the one row from another.

-          Equally the column of references are not in line with each row. This is a table that needs clearer delineation between column and rows

Table 2 and 3 suffers from the same problem

Table 4 reads well but the alignment of the last publication needs attention.

Answer: We appreciate the reviewers’ insightful suggestions. We agree with these comments. The layout of the tables has been modified as the reviewer suggested.

Figure 1:

-          White writing on a light coloured background reads very poorly. The reviewer suggests that the writing should be in black please.

-          The compilation of the figure is not comprehensible. Is there a different meaning between a circle and a rectangle?

-          Why are some lines connecting to some circles and not others?

In totality the figure is confusing. May the authors reconstruct the figure in a manner that the message is clear?

Answer: The reviewer is right. We reconstructed the figure to make it more comprehensible. In our new figure we focused on the pathogenesis of hypothyroidism and parathyroid disorder in patients with SCD and β-thalassemia. Thanks once again for your suggestions for the figure design.

General:

-          One question that the reviewer had hope would be answered is the difference in endocrinopathies between children and adolescent populations versus adults. Did the authors find any answers in the literature?

-          And when in a patient’s lifespan these endocrinopathies are most prevalent to happen?

       Answer:  We appreciate the reviewer's comment. The reviewer’s idea is very interesting. We updated our manuscript and conclusion section (Lines 441-445) accordingly as the reviewer suggested. Diabetes mellitus is more prevalent in older patients with hemoglobinopathies. Hypogonadism, growth failure, hypothyroidism, hypoparathyroidism, and adrenal insufficiency can affect children, adolescents, and children with β-thalassemia and SCD. For the development of some endocrinopathies, older age has been identified as a risk factor in various studies. However, more studies comparing the prevalence between adult and child patients with hemoglobinopathies are essential to answer these questions. The issue here is that the vast majority of the studies examined the prevalence of the endocrine disorder either in adults or children and we are not able to conclude whether the prevalence can be compared between these populations.

Reviewer 2 Report

Comments and Suggestions for Authors

This review is well-organized and covers most of the recent publications. There are minor needs to be corrected.

1. Page 7 lines 224-225. A serum ferritin level should be 2500 ng/ml, not 2.500 ng/ml.

2. Page 8 lines 285-287 Reference 125. Please check the sentence "...administration of deferoxamine could lead to delayed puberty..."

The original article concluded that "Beginning chelation treatment with deferoxamine before the age of puberty can help children with transfusion-dependent thalassemia major to attain normal sexual maturation"  

3. Figure 1 should be revised to show information, relations, and interactions among the iron and endocrine hormones. Organ dysfunctions result from the impairment of several hormones. Otherwise, figure 1 is not necessary.       

Author Response

Authors’ response

This review is well-organized and covers most of the recent publications. There are minor needs to be corrected.

Answer: We would like to thank the reviewer for reviewing our work and for the insightful comments. The reviewer’s suggestions substantially improved the quality of our work.

  1. Page 7 lines 224-225. A serum ferritin level should be 2500 ng/ml, not 2.500 ng/ml (Line 240).

Answer: Thank you for this comment. We performed the change that you suggested (Lines 300-301).

  1. Page 8 lines 285-287 Reference 125. Please check the sentence "...administration of deferoxamine could lead to delayed puberty..."

The original article concluded that "Beginning chelation treatment with deferoxamine before the age of puberty can help children with transfusion-dependent thalassemia major to attain normal sexual maturation"  

Answer: We are grateful for this comment. We made the essential changes in the manuscript (Lines 300-301). 

  1. Figure 1 should be revised to show information, relations, and interactions among the iron and endocrine hormones. Organ dysfunctions result from the impairment of several hormones. Otherwise, figure 1 is not necessary.     

Answer: We want to thank the reviewer for this comment. We modified our figure to make it more comprehensible. In the revised version of our manuscript, figure 1 focuses on the pathogenesis of hypothyroidism and parathyroid dysfunction in patients with hemoglobinopathies.

Round 2

Reviewer 1 Report

Comments and Suggestions for Authors

The reviewer thanks the authors for their efforts in improving the manuscript.